# Aggression, Boldness, and Exploration Personality Traits in the Subterranean Naked Mole-Rat (*Heterocephalus glaber*) Disperser Morphs

**DOI:** 10.3390/ani12223083

**Published:** 2022-11-09

**Authors:** Tshepiso Lesedi Majelantle, Andre Ganswindt, Christian Walter Werner Pirk, Nigel Charles Bennett, Daniel William Hart

**Affiliations:** 1Mammal Research Institute, Department of Zoology and Entomology, University of Pretoria, Pretoria 0028, South Africa; 2Social Insects Research Group, Department of Zoology & Entomology, University of Pretoria, Pretoria 0028, South Africa

**Keywords:** African mole-rats, animal personality, barricade, Bathyergidae, behavioral acts, behavioral syndrome, response to disturbance

## Abstract

**Simple Summary:**

Naked mole-rats have a unique social system similar to insects, where there is one reproductive female, one to three reproductive males, and the rest are non-reproductive members of the colony. Within the non-reproductive members, there are dispersers that attempt to leave the colony. We aimed to investigate if naked mole-rat dispersers have consistent behavioral responses to situations and therefore possess animal personality traits. The naked mole-rat dispersers showed consistent responses to different situations, and had consistent differences between individuals. In addition, we recorded how naked mole-rats respond to situations that threaten their survival, such as a new environment and predators. Further investigation into these animal personality traits and how they compare between other colony members, such as workers and soldiers are needed to understand the naked mole-rat social system.

**Abstract:**

Animal personality traits (consistent behavioral differences between individuals in their behavior across time and/or situation) affect individual fitness through facets, such as dispersal. In eusocial naked mole-rat (*Heterocephalus glaber*) colonies, a disperser morph may arise with distinct morphological, behavioral, and physiological characteristics. This study aimed to quantify the personality traits of a cohort of disperser morphs of naked mole-rat (NMR). Behavioral tests were performed on twelve disperser morphs (six males and six females) in an observation tunnel system that was novel and unfamiliar. Novel stimuli (fresh snakeskin, tissue paper, and conspecific of the same sex) were introduced for fifteen minutes, and the behavioral acts of the individual were recorded. A total of 30 behaviors were noted during the behavioral tests of which eight were used to quantify aggression, boldness, and exploration. The NMR disperser morphs showed consistent individual differences in boldness, and exploration across time and test, indicating a distinct personality. In addition, new naked mole-rat responses including disturbance behaviors; confront, barricade, and stay-away, were recorded. Further investigations into the relationships between animal personality traits and social hierarchy position in entire colonies are needed for more informative results as we further investigate the role of personality in cooperatively breeding societies.

## 1. Introduction

Animal personality traits are consistent behaviors within individuals and consistent differences between individuals across time and/or situation [1]. Basic animal personality traits include behavioral traits such as: activity (the propensity to move through a landscape), aggression (the propensity to exhibit antagonistic behaviors towards conspecifics), boldness (the propensity to respond to a situation which potentially threatens survival), exploration (the propensity to be active and collect information in a novel situation), and sociability (the propensity to interact with conspecifics) [2,3,4]. In addition, there are complex animal personality traits which include behavioral traits such as: agreeableness, assertiveness, conscientiousness, inquisitiveness, and patience [5,6]. Overall, animal personality traits are quantified for a wide range of species including arthropods [7], birds [8], fish [1], amphibians [9], reptiles [10], mammals [11], and non-human primates [12] to cite several examples. There are two main types of animal personality assessment, namely, behavioral codings and observer trait ratings [13]. Behavioral coding requires measuring the frequency of behavior from a well-defined and discrete ethogram, whereas observer trait ratings require an observer to rate behavior based on a pre-defined questionnaire [14]. Thus, there is no universal methodology to test for animal personality traits between various taxa primarily due to differences in their behavioral repertoires and complexity, life-history traits, and observer access to the species [13,15,16]. Thus, it is vital to validate and investigate species-specific animal personality tests [17]. Animal personality traits are important since they directly affect fitness through attractiveness to mates, interspecific competition, and reaction to predators [3]. In social animals, personality traits possibly underpin factors such as social interactions, group dynamics, and dispersal [18]. The correlation between multiple animal personality traits across time, situation or both is defined as a behavioral syndrome [1]. For examples individuals with a proactive behavioral syndrome exhibit heightened aggressiveness, a higher degree of exploration and greater boldness, whereas reactive individuals are characterized by a lower degree of aggressiveness, reduced exploration, and low boldness [19].

The naked mole-rat (*Heterocephalus glaber*, [20]) is a subterranean mammal which is one of the only two known species of mammals classified as eusocial [21,22,23]. Whereby eusociality is defined as a social system with division of labor, overlap in generations, cooperative care of young, and lifetime reproductive success of individuals within a colony is less than 1% [24,25,26]. The naked mole-rats occur in large colonies which may contain between 60 to 80 individuals with only a single breeding female (queen), one to three breeding male consorts, and all other individuals being non-reproductive, subordinate and reproductively suppressed [22,27,28,29]. Within colonies, there is a reproductive division of labor whereby non-reproductive subordinates perform tasks, such as burrow maintenance, foraging, and colony defense, while reproductive animals are primarily responsible for procreation [22,30,31]. In addition, in naked mole-rat colonies, there exists the occasional disperser morph, which has greater total body fat, is physiologically reproductively active, and has a tendency to attempt to leave the colony [32], possibly arising due to intrinsic colony factors such as queen temperament and colony size [33].

There is variability in the expression of behaviors within naked mole-rat colonies that may play an important role in naked mole-rat reproductive suppression, colony maintenance, and social hierarchy [30,34]. For example, antagonistic behaviors, such as shoving (vigorous, prolonged head-to-head pushes between individuals), are common in a naked mole-rat colony [30,35]. Evidence thus far suggests that the queen is the most aggressive colony member, followed by breeding males, and to a lesser extent some of the larger non-breeding females, especially females born to early litters which exhibit social queueing for feeding (hierarchal order for opportunities) [30,33]. The non-reproductive division of labor is linked to differences in behavior, which has led to the description of three behavioral phenotypes: soldiers, dispersers, and workers, however, further investigation is required to determine if the behavioral phenotypes are discrete or continuous [36]. The explorative behavior of burrow openings is linked to disperser morphs that persistently attempt to disperse [32]. Soldiers are linked to colony defense [22,37], which is possibly determined by the aggressive behavior towards foreign conspecifics [33]. Finally, workers are non-aggressive and non-explorative individuals [33,38].

Herein, disperser morph naked mole-rats were selected to establish a method using behavior codings to investigate animal personality traits: aggression, boldness and exploration, in naked mole-rats. This behavioral phenotype was selected because it is the most likely to exhibit boldness, exploration, and a degree of aggression. Thus, the aims were to; (1) to establish behavioral tests which allow for the quantification of animal personality traits (aggression, boldness, and exploration) in the naked mole-rat disperser morphs; (2) examine if within- and between- individual variation in naked mole-rat disperser morph personality traits is repeatable across time and context; (3) examine if these repeatable personality traits are correlated forming a behavioral syndrome. We hypothesize that within naked mole-rat disperser morphs there will be variable personalities as indicated by consistent differences in their behavior across time.

## 2. Materials and Methods

### 2.1. Study Animals

Behavioral tests were performed on twelve adult disperser morph naked mole-rats (six male and six females) with a body mass of 41.2 ± 0.84 g (mean ± SE, range = 31.4–54.4 g) between September 2020 and April 2021. The animals were healthy, captive-bred and housed at the University of Pretoria. The room was maintained on a 12L:12D light schedule and a temperature and humidity range of between 29–30 °C and 40–60%, respectively. Disperser morphs were kept alone in semi-clear individual plastic chambers (length = 35 cm, width = 30 cm, and height = 20 cm) with nesting material comprised of sterilized wood shavings. These animals were previously separated for 598 ± 299 days (range = 119–897 days) at the beginning of their personality assay as disperser morphs based on body mass and the frequency of escape attempts from the natal colony [31]. The animals were provided with an *ad libitum* amount of chopped vegetables, mainly sweet potato (*Ipomoea batatas*), cucumber (*Cucumis sativus*), apples (*Malus domestica*), and bell pepper (*Capsicum annum*) for feeding.

### 2.2. Personality Assay

The various types of behavioral acts are described in detail in Table 1 and were quantified by introducing each subject to a novel Perspex tunnel system of a specified diameter and length (Figure 1). The novel tunnel system included an acclimation pod with nesting material from the subject’s enclosure (familiar scent), connected to two other pods (a control pod with unscented wood shavings, and a novel object/experimental pod) with two tunnels (Figure 1). Each subject was weighed prior to being placed into the acclimation pod with closed tunnel entrances. The animal was allowed to acclimate to the pod for a period of 10 min. Thereafter, the tunnels were opened and the naked mole-rat was allowed to emerge, and all behavioral acts (Table 1) were recorded for 15 min by video cameras. The animal was considered to have entered the tunnel/pod when all four limbs were inside. After 15 min, the doors to the tunnel system were closed, and if the subject was not in the acclimation pod, the doors were closed after the mole-rat’s return. The subject was given a further 10 min to rest. A new stimulus object was added to the novel object pod, and the doors of the acclimation pod were opened once again. The time of emergence of the naked mole-rat and all behavioral acts were recorded for 15 min. Thereafter, the animal was allowed to rest in the closed acclimation pod for a further 10 min. This procedure was repeated with three novel objects: a 20 cm piece of fresh snakeskin from a mole snake (*Pseudaspis cana*) to elicit a predator response (boldness test), 20 cm piece of tissue paper to validate the snakeskin response (boldness validation test), and finally a conspecific of the same sex (aggression test). During the conspecific test, the entrance into the novel object pod was blocked by a wire mesh. Thus, the animals could smell and hear each other, but did not have physical contact. The personality tests were repeated in the same order, on the same individual after 1-, 3-, 5-, and 7- day(s) from the previous test and all began at the same time (09:00). Behavioral acts performed by the mole-rats are described by Lacey et al. [35], in addition to some new behavioral acts are described in Table 1. All behavioral acts were recorded using MS Excel™ by three observers, including the principal investigator. The PI scored the behavioral recordings from the first individual for each test scored by the two other observers to control for consistency. Speed was quantified using the time taken for the animal to reach known distances within the novel tunnel system. For two test days for two individuals (one male, one female), there were camera failures and during one boldness validation test (same male) there was a power outage, and therefore this data was excluded from analysis. Thus, a total of 58 exploration tests, 58 boldness tests, 57 boldness validation tests, and 58 aggression tests were conducted on 12 individuals 0, 1, 3,5,7- days after the previous test.

### 2.3. Data Analysis

All analyses were performed using R Statistical Software [39]. All recorded behavioral acts were converted to a proportion of the total time of the test (900 s). A nomination approach based on our perception of the relevant response of behavioral acts to the stimulus was used to select the behavioral acts of biological relevance to the species (Table 1). The selected behaviors were considered biologically relevant because they were commonly performed by the naked mole-rats, occurred during all personality tests by most individuals, and could not have occurred in response to a different stimulus (Table 1). Principal Component Analysis (PCA) was used on selected behavioral acts (including speed) for each individual during each test to quantify personality, and the final personality value was from the first PCA (Table 2). Pearson’s correlation test was used to test for consistency between personality tests. Normality was tested for each personality test day using the Shapiro Wilk test. A Holm-Bonferroni correction was done to the correlation tests applied to each personality test. In addition, a Wilcoxon matched pairs signed rank test was used to test the difference between the boldness test and boldness validation test to validate the effectiveness of the snakeskin. An Analysis of Variance was used to test if there was a significant difference between individual PCA values. A linear mixed effects model via lme4 package [40] with individual animal ID as random factor was used to test if body mass and sex had a significant effect on personality score calculated by the corresponding personality trait, for the aggression test, the natal colony of the conspecific was included as a predictor. Normality was tested for visually using the QQ plot and Levene’s test on model residuals.

The confront, barrier, and avoid response to disturbance behavioral acts (see results section; Table 2) were calculated for each individual during the boldness and exploration tests using the following formulas (where AN = Acclimation-Novel tunnel, Figure 1):Confront = ∑time spent in Novel pod
Barricade= ∑time spent peeking in AN + ∑time digging in AN + ∑time crouch advancing in AN + ∑time freezing in AN
Stay away = 900 − (∑time spent in Novel pod + (∑time spent in AN − Barricade))

A Pearson’s correlation test was used to test if there was a significant correlation between the boldness or exploration tests, and the response to disturbance behavioral acts. A Holm-Bonferroni correction was performed for all correlation tests applied for response to disturbance behavioral acts.

Finally, behavioral syndromes were investigated using Pearson’s correlation between PCA values from each personality test. Thereafter, a multivariate Markov chain Monte Carlo generalized linear mixed model via MCMCglmm package [41] with animal ID as a random effect was used to test if body mass, sex, and test repetition had a significant effect on the correlated personality tests. Significant differences were checked using a Wald test on posterior means and variances. For all statistical tests, the level of significance was *p* ≤ 0.05. All tests met the normality assumptions.

## 3. Results

In total, 24 discrete behavioral acts were observed during the behavioral tests (Table 1). However, only seven behavioral acts were used for the exploration, boldness, and aggression PCAs (Table 1). The variance explained by PCA for the exploration test, boldness test, boldness validation, and aggression were 35.1%, 27.1%, 32.3%, and 28.4%, respectively (Table 2). Avoidance/shy/docile behaviors were positively loaded in the exploration, boldness, boldness validation test (except gnawing), and aggression test (except retreat) (Table 2). Thus, PCA values less than 0 were considered explorative, bold, and aggressive individuals.

### 3.1. Aggression

Results from the first aggression test were not significantly correlated with results from 1 (*p* = 0.078; correlation coefficient = 0.637), 3 (*p* = 0.063; correlation coefficient = 0.676), 5 (*p* = 0.078; correlation coefficient = 0.601), and 7 days (*p* = 0.907; correlation coefficient = −0.042). Thus, the aggression behavior was not consistent within an individual over time. There were significant differences between individual naked mole-rat PCA scores (*p* < 0.001, F_11,46_ = 6.987, DF = 11, R^2^ = 63%, Figure 2), thus showing consistent differences between individuals. The within individual average variance was 1.073, and between individual variance was 7.496. The PCA values had no significant relationship with body mass (*p* = 0.603, χ = 2.271, DF = 1, Figure 2). There was no significant difference in aggression PCA values between the sexes (*p* = 0.841, χ = 0.040, DF = 1, Figure 2) and between conspecifics natal colonies (*p* = 0.452, χ = 3.671, DF = 4).

### 3.2. Boldness

There was no significant difference/location shift (*p* = 0.731, W = 1715) between PCA score during the boldness test, and the boldness validation test. Results from the first boldness test, were significantly positively correlated with results from 1 (*p* = 0.010; correlation coefficient = 0.771), 3 (*p* = 0.002; correlation coefficient = 0.864), and 5 (*p* = 0.036; correlation coefficient = 0.666) days apart from previous boldness tests. However, there was no significant correlation between the first boldness test and the test 7 days (*p* = 0.104; correlation coefficient = 0.549) after the previous test. Therefore, the behavioral acts were consistent over time (for the first four repetitions), but were inconsistent 7 days after the previous test (fifth repetition). Results from the first boldness validation test were significantly positively correlated with results from 1 (*p* < 0.001; correlation coefficient = 0.903), 3 (*p* = 0.011; correlation coefficient = 0.765), 5 (*p* = 0.011; correlation coefficient = 0.790), 7 (*p* = 0.008; correlation coefficient = 0.779) days apart from the previous boldness validation tests. All further analyses were conducted using data from the boldness validation because there was no difference between the two tests, and the boldness validation had a higher correlation coefficient and proportion of variance explained by the PCA (Table 2). There were significant differences between individual naked mole-rat PCA scores (*p* = <0.001, F_11,45_ = 14.618, DF = 11, R^2^ = 73%, Figure 2) thus showing consistent differences between individuals. Within individual average variance was 0.717, and between individual variance was 10.475. There was no significant relationship between PCA values and body mass (*p* = 0.241, χ = 1.376, DF = 1, Figure 2). Males had significantly higher (greater shyness) PCA values than females (*p* = 0.011, χ = 6.521, DF = 1, Figure 2).

### 3.3. Exploration

Results from the first test, were significantly positively correlated with results from 1 (*p* = 0.020; correlation coefficient = 0.657), 3 (*p* = 0.013; correlation coefficient = 0.734), 5 (*p* = 0.010; correlation coefficient = 0.766), and 7 (*p* = 0.010; correlation coefficient = 0.839) days apart from previous exploration test. Thus, showing consistency in exploration across time. There were significant differences between individual naked mole rat PCA scores (*p* < 0.001, F_11,46_ = 27.125, DF = 11, R^2^ = 83%, Figure 2) thus showing consistent differences between individuals. Within individual average variance was 0.590, and between individual variance was 12.355. There was no significant relationship between PCA values and the body mass of the animal (*p* = 0.856, χ = 0.033, DF = 1). In addition, there was no significant difference between the sexes (*p* = 0.188, χ = 1.737, DF = 1).

Three responses to disturbance behaviors were recorded during the boldness test (Table 3). There was a significant negative correlation between boldness (*p* = 0.012; correlation coefficient = −0.400) or exploration (*p* = 0.006; correlation coefficient = −0.423) and the confront behavior (Figure 3). Thus, bold/explorative individuals were more likely to confront a novel situation or perceived threat. In addition, there was a significant positive correlation between boldness (*p* = 0.013; correlation coefficient = 0.397) or exploration (*p* < 0.001; correlation coefficient = 0.514) and the barricade behavior (Figure 3), Thus, shy/avoidant individuals were more likely to attempt to barricade a novel situation or perceived threat. There was no significant correlation (*p* = 1.000) between boldness (correlation coefficient = 0.123) or exploration (correlation coefficient = 0.085) and the stay away behavior. This provides preliminary evidence for barricade as a response to disturbance (i.e., perceived threats) behavior in naked mole-rats.

### 3.4. Behavioral Syndrome

The PCA values from the boldness and aggression test (correlation coefficient = 0.517), and the aggression and exploration (correlation coefficient = 0.502) test were significantly positively correlated (*p* < 0.001). However, the correlation was not considered a behavioral syndrome due to the low correlation coefficients between variables and no significant consistency within aggression. The PCA values from the boldness and exploration test were significantly positively correlated (*p* < 0.001, correlation coefficient = 0.809). Therefore, an exploration-boldness behavioral syndrome exists in naked mole-rat disperser morphs (Figure 4). There was no significant difference between sexes (*p* = 0.166) and between 0, 1, 3,5, and 7 days after the previous test for repetition of behavioral tests (*p* = 0.240) in the exploration-boldness. In addition, there was no significant relationship between body mass and exploration-boldness (*p* = 0.671).

## 4. Discussion

The results from this study illustrate the successful application of behavioral tests which allow the quantification of animal personality traits in the subterranean naked mole-rat disperser morph. The aggression test had no statistically significant consistent behavioral acts within individuals across time and frequency. Whereas the boldness and exploration test showed consistent behavioral acts within individuals across time and frequency. However, day 7 (five repetitions), behavioral acts were not consistent during the boldness snakeskin test. During the boldness tests, the animals showed no difference in reaction to the snakeskin or to the tissue paper. In addition, boldness was the only test to show statistically significant sexual differences in animal personality traits. Analysis of the behavioral data allowed the description of new behavioral acts in the naked mole-rat, and the identification of one response to disturbance behavior, namely barricade behavior, in naked mole-rats which is potentially unique to subterranean mammals. Finally, the results show a boldness-exploration behavioral syndrome in naked mole-rat disperser morphs.

In this study, aggression in the naked mole-rat disperser morph was not significantly different between the sexes or with body size. In addition, these behavioral acts were not consistent over time, thus aggression appears to not be a personality trait of naked mole-rat dispersers. This result is not completely unexpected since Toor et al. [38] reported that dispersers preferred unfamiliar individuals compared to the workers, but exhibited lower aggression towards novel animals compared to a soldier. Previous studies have used shoving behavior [35] as a proxy for aggression, and this behavior was strongly associated with individual dominance and reproductive status [34]. Specifically, aggressive behavior was directed by high-ranking individuals, particularly the queen towards individuals that posed the greatest threat to her position of dominance, and vice versa [34]. However, these aggressive behaviors are mostly likely not directed towards disperser morphs [34].

Naked mole-rat xenophobic behavior (aggressive inter-colony behavior) is well-documented, with olfaction being the main contributor to colony member recognition [37] and more recently vocalization [42]. These xenophobic behaviors are likely due to competition between colonies for resources, or the kidnapping of pups for subordinate/slavery roles in the colony [43]. In previous studies, disperser morphs were less likely to display aggressive behaviors towards a conspecific from a different colony as compared to their non-disperser counterparts [32,33]. This docile behavior towards foreign conspecifics may well be linked to success in being accepted in a foreign colony, and subsequently reproductive success. Alternatively, the experimental setup, and behaviors selected to indicate aggression were not sufficient to detect consistent responses within individuals since the aggression test had the highest within individual variation compared to the exploration and boldness test. Dyadic encounters in a tunnel shaped arena may be a more useful approach [44].

Some studies apply predator cues [45], novel arena emergence [46], or novel objects such as toys [47] to test for boldness. Given the poor vision in naked mole-rats [48] and the importance of olfaction in their recognition behavior [34], the current study applied both olfactory predator cues [49] by using fresh snakeskin [21], and tissue paper, a novel object. There was no difference between the snake-skin test and the tissue paper test, which suggests that there was no difference in how the animal perceived the two different stimuli. One possible reason for this finding is that mole snake distribution is limited to southern Africa, whereas naked mole-rats naturally occur in eastern Africa. Consequently, the animals did not recognize the scent as a threat. Thus, both the snakeskin and tissue paper were potentially identifiable as the same novel object, or it could have decided that the snake and tissue both had a foreign odor. Alternatively, the fresh snakeskin scent could not have been limited to the novel pod, but moreover the entire novel tunnel system (or experimental room), thus preventing the animal from responding to the scent as a predator cue in the novel pod. The results suggest that female naked mole-rat disperser morphs are bolder compared to their male counterparts. This supports previous evidence that shows that male dispersers are less active than non-dispersers [32,33]. The lower activity could be linked to the animal’s propensity to explore in a risky situation. In addition, boldness could be selected for in female disperser morphs specifically due to them having to enter colonies with aggressive queens. Thus, boldness may contribute to the reproductive success and survival of the female after leaving the natal colony.

It is common for rodents to either attack disturbances or avoid/retreat from where the disturbance has been detected. The barricade response to disturbance has not yet been described in the naked mole-rat. Similar barricading behavior has been observed in the solitary free-living silvery mole-rats (*Heliophobius argenteocinereus*), whereby individuals separated themselves from the disturbance by using plugs of soil [50]. The naked mole-rats could have been attempting to dig or create a ‘soil’ plug, but were unable to do so due to the experimental set-up (plastic tubes instead of sandy tunnels). Interestingly, the results here suggest that barricading behavior is more likely performed by shy and avoidant individuals. The barricading behavior could possibly be limited to disperser morphs alone, or it may also arise in shy and avoidant soldiers in naked mole-rat colonies.

Naked mole-rat disperser morphs are usually the more explorative members of the colony, since they have the propensity to find and exit from openings in the tunnel system to disperse into adjacent colonies [32,33]. These animals have significantly different physiological characteristics (higher levels of bioactive plasma luteinizing hormones) and morphological characteristics (higher mass and greater body fat content) compared to non-reproductive individuals that are thought to assist with their dispersal [32]. However, the results here included disperser morphs with an avoidance personality trait. Disperser morphs are usually the explorative individuals of the colony [33]. Since the personality trait is probably continuous, there is a possibility that when compared to the rest of the colony, the dispersers occupy the explorative end of the spectrum; thus, compared to other dispersers, these individuals are avoidant, but possibly compared to other members of the colony, they are explorative.

Evidence presented here shows the existence of an exploration-boldness behavioral syndrome in naked mole-rat disperser morphs. The personality traits and behavioral syndrome shown here suggest that the previously described disperser behavioral phenotype in naked mole-rats [36] is continuous instead of discrete. Animal personality traits were previously quantified in the cooperatively breeding Ansell’s mole-rat (*Fukomys anselli*) non-breeders [51], thus, it is likely animal personality traits are present in other naked mole-rat non-breeders. If so, since the boldness-exploration axis is continuous, the disperser morphs probably occupy a specific range, and other behavioral phenotypes occupy different ranges within the spectrum. Alternatively, isolation from other members of a colony is potentially stressful to naked mole-rats [52,53]. Social isolation could have physiological and behavioral consequences [54], such as increased aggression, avoidant behaviors, and reduced boldness [55]. Since the animals used here have been isolated long-term, what started as a discrete behavioral phenotype could have changed into a continuation between individual variation as a consequence of isolation. On the other hand, identifying disperser morphs following the methods of O’Riain et al. [32] could potentially select individuals not limited to the disperser behavioral phenotype. Further studies should compare the personality traits of dispersers identified by the two methods (O’Riain et al. [32] and Toor et al. [33]), to track if there are changes in behavior after isolation and compare these dispersers to other non-disperser behavioral phenotypes within a colony.

## 5. Conclusions

This study established a successful method for quantifying animal personality traits of boldness and exploration in naked mole-rat disperser morphs which are consistent between time and frequency. However, the method was unsuccessful in quantifying consistent differences in the aggression trait. Furthermore, there was correlation between the personality traits boldness and exploration, which suggests the existence of behavioral syndromes within the species. In addition, there were significant sexual differences in boldness behavior, likely due to the influence of the queen in natal colonies. Explorative behavior is a well-known trait in disperser morphs, however, individuals showing comparatively high avoidance behaviors may do so because the trait is continuous. Finally, barricade behavior was described as a potential response to disturbance in the species. In colony settings, the observed personality traits and response to disturbance behavior is possibly linked to social hierarchy dynamics and dispersal events.

## Figures and Tables

**Figure 1 animals-12-03083-f001:**
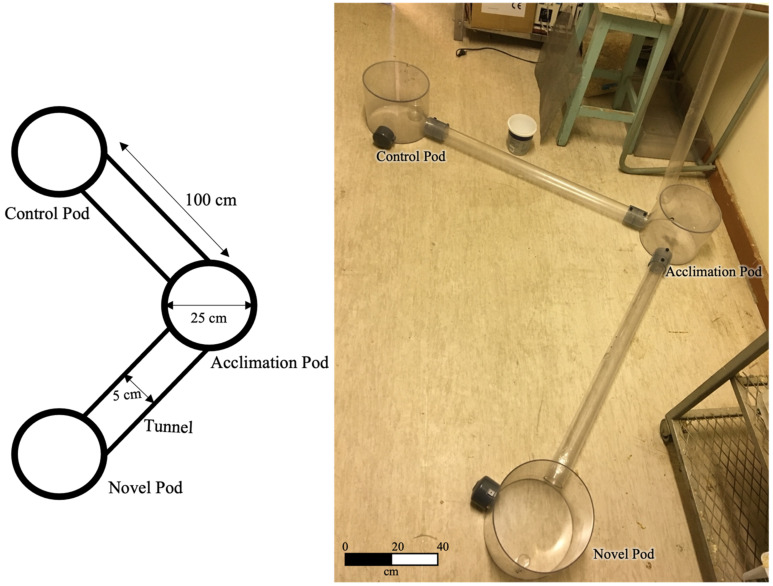
Drawing and photograph of the set-up of observational tunnels as a novel tunnel system for naked mole-rat behavioral tests.

**Figure 2 animals-12-03083-f002:**
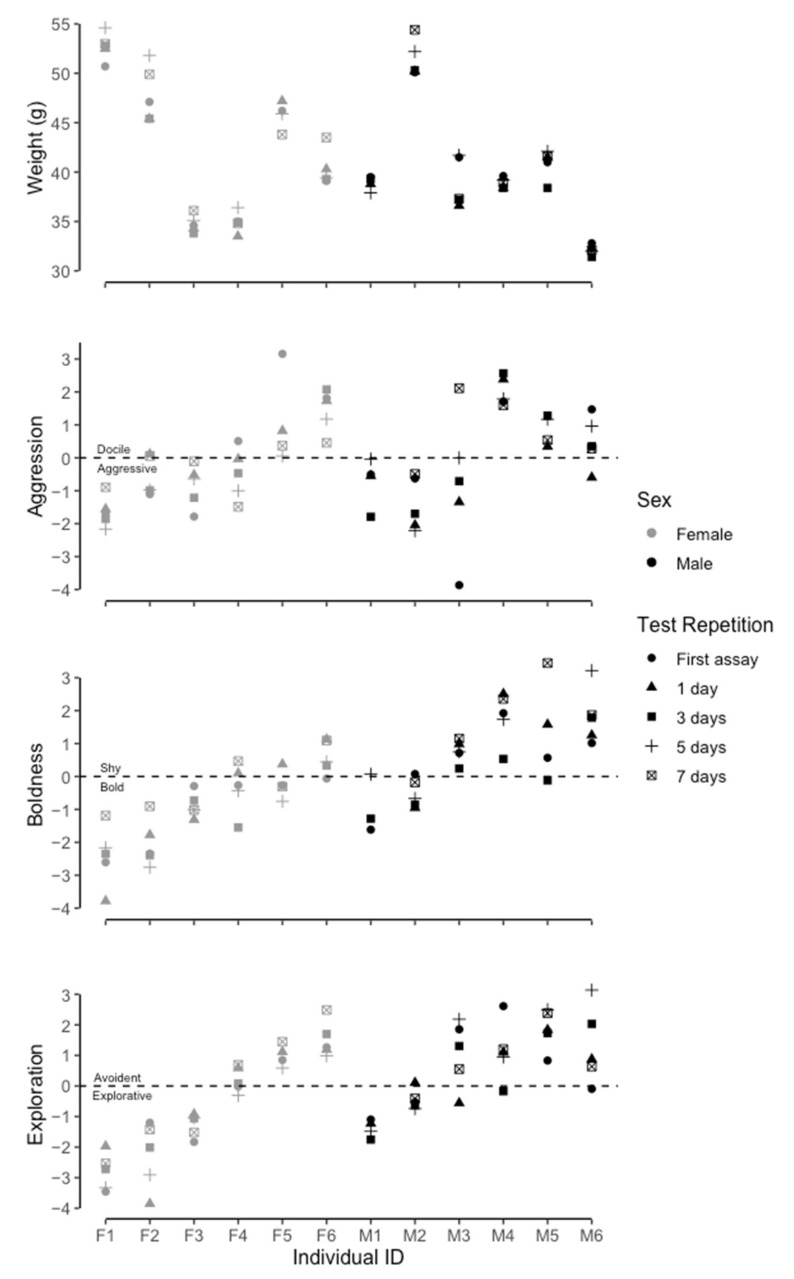
Between and within individual variation in body mass (g), and PCA values from Aggression, Boldness and Exploration tests, respectively.

**Figure 3 animals-12-03083-f003:**
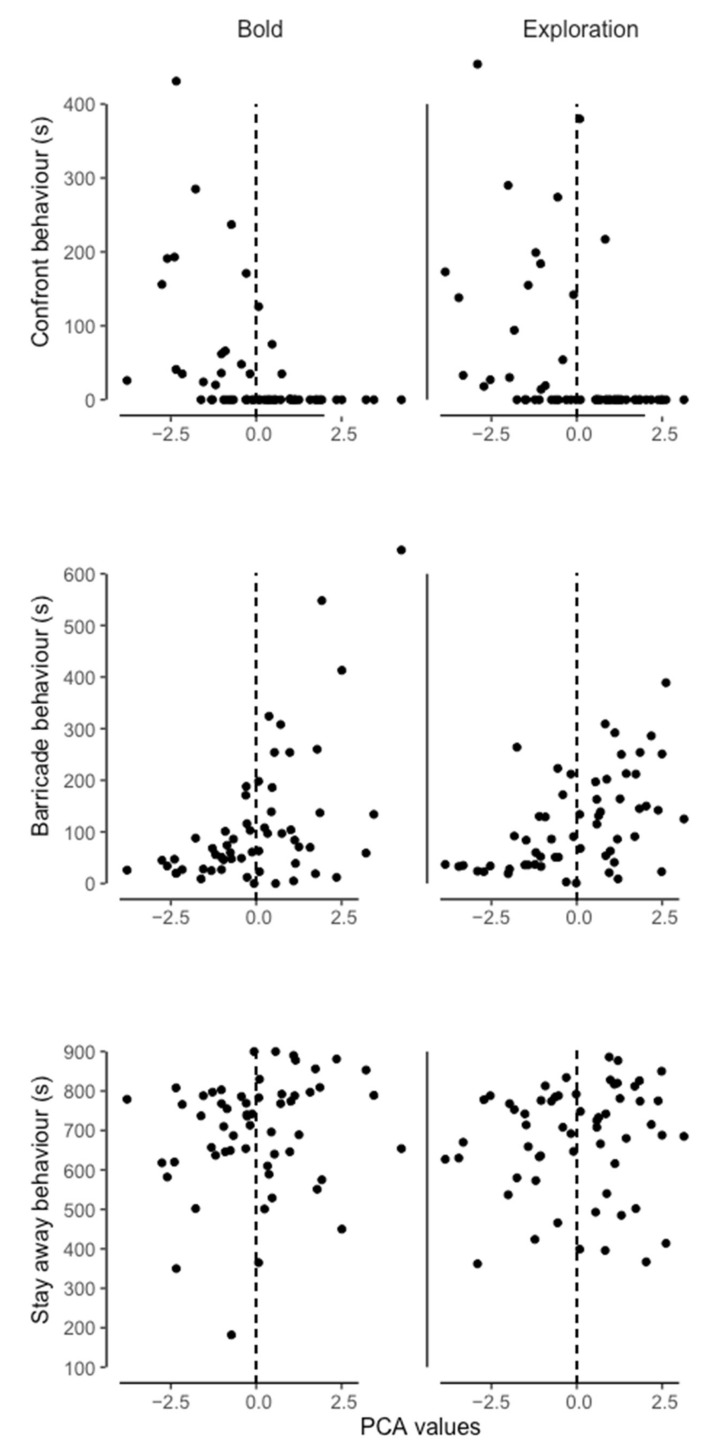
Values for each response to disturbance behaviors for naked mole-rat disperser individuals during boldness and exploration tests. Bold and explorative individuals had negative PCA values, whereas shy and avoidant individuals have positive PCA values.

**Figure 4 animals-12-03083-f004:**
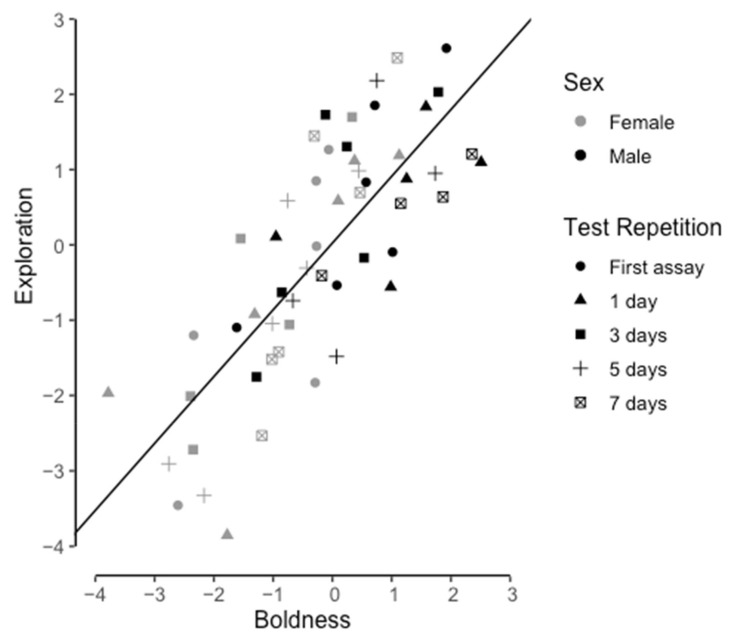
Positive correlation (correlation coefficient r = 0.809) between PCA values from Boldness and Exploration tests.

**Table 1 animals-12-03083-t001:** An ethogram for observed naked mole-rat behavioral acts described by Lacey et al. [35] during personality assays.

Title	Description
Climbing *	The subject stands on its hind legs and uses its forefeet alternately to scratch the walls of the pod.
Crouch advancing *	Crouching—The subject stands in a hunched posture, with extremities held under the body (individuals often shiver while crouching).
	Crouch advancing—The subject slowly moves forward a few steps with its legs bent and its body held close to the substrate; the subject pauses, and then moves forward again (usually accompanied by sniffing).
Darting	Rapidly jumps a short distance (one or two body lengths) forward.
Defaecating	The subject expels feces, and usually does anal sniffing before expelling feces
Digging	Combination of foreleg digging, nose shoveling and/or sweeping behaviors on a specific area
	Foreleg digging—One foreleg is brought forward, and the foot is scraped along a surface then returned to a position under the body. Action is repeated rapidly using alternate forelegs. Often occurs in succession with gnawing.
	Nose shoveling—The subject buries its entire nose and part of face in sawdust and pushes the saw dust away with face.
	Sweeping—Subject kicks loose items behind itself while moving backward
**Entrance**	**The subject enters the pods or tunnel when both fore and hind legs are in the pod/tunnel**
Fall	The subject falls on back and struggles to get back on feet
Freezing *	The subject suddenly ceases all activity, holding itself in the same position it had been in at the time of the disturbance/encounter
Gnawing *	The subject’s incisors are scraped along a surface of an object (wire mesh in Aggression test)
Grooming	The subject wipes/scratches the face, flank, or underarm with feet.
	Anal sniffing—The subject doubles over so that it is sitting on its hindquarters with its mouth and nose in contact with its anus.
Ignore	The subject moves near or past the stimulus object without displaying any behavioral response to the object
Mouth carrying	The subject holds an item between the incisors and lifts it off the tunnel floor carrying it whilst moving forward or backwards.
Nibbling	The subject holds food item with both forepaws and consumes small portions of food using the tongue and incisors.
	Auto coprophagy -The subject consumes its own feces.
**Peeking ***	**Subject inserts head whilst sniffing only into tunnel or pod before entrance or retreat.**
Reclining	The subject lies motionless on its side, back, or stomach with eyes closed.
**Retreat ***	**Abrupt movement backward in response to a disturbance/novel encounter. This could take place after freezing or movement**
**Rotation ***	**The subject moves along the edges of an open space (pod) alternating between clockwise and anticlockwise whilst sniffing.**
**Rubbing**	**The subject presses its body against the walls of the tunnel or pod whilst moving forward or backwards.**
Shivering	The subject rapidly shakes or quivers its body and extremities
Sniffing	Sharp and distinct nose movements whilst slowly nodding head on or close to an object
Travelling	General movement from one point to another
	Turn around—The subject reverses the direction of movement either by completing a forward somersault followed by a 180° twist or completing a turn.
Urinating	A subject excretes, usually with hind legs splayed and scratching immediately after.
Vocalization	The subject vocalizes in response to the stimulus

Note: Bold = newly described behavioral acts, * behavioural acts used for aggression, epxloration and boldness PCAs.

**Table 2 animals-12-03083-t002:** First principal component analysis loadings for each behavior for each personality test.

Behavior	Exploration Test	Boldness Test	Boldness Validation Test	Aggression Test
Climbing	−0.395	−0.444	−0.407	−0.038
Gnawing	0.135	*0.071*	*0.053*	−0.444
Rotation	−0.463	−0.477	−0.456	*0.029*
Speed	−0.384	−0.381	−0.267	−0.370
**Crouch advancing**	**0.501**	**0.563**	**0.534**	**0.545**
**Peeking**	**0.250**	**0.282**	**0.151**	**0.478**
**Freezing**	**0.218**	**0.131**	**0.436**	**0.181**
**Retreat**	**0.321**	**0.110**	**0.234**	**−0.323**
Standard deviation	1.677	1.471	1.608	1.507
Proportion of Variance	0.351	0.271	0.323	0.284

Note: Bold = avoidance/shy/docile behaviors, Italics = loadings < 0.1.

**Table 3 animals-12-03083-t003:** Description of observed naked mole-rat response to disturbance behaviors during boldness and aggression tests.

Anti-Predator Strategy	Description
Confront	Subject enters novel pod and sniffs and/or gnaws at snakeskin or gate blocking conspecific
**Barricade**	**Subject blocks entrance into the tunnel from the novel pod by peeking for a long period of time without entering the novel pod or digs in the acclimation-novel tunnel**
Stay away	Subject avoids novel pod and/or acclimation-novel pod tunnel

Note: Bold = newly described anti-predator strategy.

## Data Availability

Data supporting the reported results will be sent by corresponding author upon request.

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
