# Peer review of "Aggression, Boldness, and Exploration Personality Traits in the Subterranean Naked Mole-Rat (Heterocephalus glaber) Disperser Morphs"

_animals, 2022, doi:10.3390/ani12223083_

Round 1

Reviewer 1 Report

The manuscript of Majelantle and colleagues has major impact on the overall ‘hot’ question of personality in animals. As discussed in the recent month/year in laboratory rodents, here a really nice example of a study with naked mole rats is presented. And indeed, personality traits or even syndrome can be defined, with the help of newly deciphered and nicely described behavioral acts.

Although the manuscript is well written and all results are nicely presented, I could find some issues which , if changed, would make the manuscript even more readable and I suggest to do so in a minor revision.

1.      Please combine figure 1 and s1.

2.      Boldness test, boldness validation test and aggression test are nicely described, but the name of the tests are missing in the method section, so the reader looses a bit the overview, so please add the ‘clear‘ name, e.g. in brackets, in line 129-131

3.      I would remove commas in line 195, 213 and 232

4.      For the boldness validation test: was cotton or paper tissue used? This seems to be a bit mixed up, for me these are difference materials

5.      With 12 animals and 5 tests each, this should be result in 60 single data points, but you report only 58. What happened for the last 2?

6.      Bodyweight data: F1 seems to drop in weight at test day 5 for almost 10g. The other animals seem to be stable, so what happened here?

7.      Is the conspecific for sure not a threat? How is this made sure?

8.      As you discuss aggression as a trait of natal colony ‘culture’… How many colonies do your animals descend from and have you controlled your data about that? That would be a discussion of the experimental unit you were using, I assume.

9.      I am not very happy about your discussion about the boldness and boldness validation test and the conclusion, that the knowledge of predator smell was kind of lost. These instincts are normally quite well preserved as e.g. in lab rodents (being bred for almost 80 years now in captivity) the behavior towards fox smell. Although I trust your results and the conclusions you draw, but perhaps some ‘fresh’ predator smell would have been better. Or the smell was so intense that it was already in the whole room…

10.  As you discuss the previous live in the original colony, how long were your NMR already separated? Did this make any effect on their behavior?

11.  Some more details about hosing conditions would be more than useful I recommend the ARRIVE guidelines checklist to consider some forgotten details, e.g. health status, housing conditions like cage sizes etc.

12.  Some basic data would be nice as you measured e.g. aggression: Perhaps a supplementary table or so would be nice to get some ‘contact’ of the basic data – e.g. what a aggressive NMR means in behavioral acts.

Reviewer 2 Report

Dear Authors,

Thank you for submitting this paper that explores the behavior of disperser morph naked mole rats in response to novel object and aggression tests. This an interesting area with a lot of potential future directions.

1. At current however, there seem to be some large revisions required in the manuscript to ensure the work is scientifically robust. I have attached the PDF version of the manuscript with specific comments. Additionally, please consider the following points: 

1. Introduction. The explanation of animal personality research is general and vague, with sweeping generalisations. Much greater depth is needed. A clearer explanation is needed on the physiology of the disperser mole rats - otherwise this work is not repeatable.

2. Length of time that individuals were separated from the colony. The amount of time elapsed since animals were separated is highly likely to influence their behaviour, especially when encountering other individuals. This needs to be explained clearly.

3. Statistics. Please provide actual p values rather than P<0.05. You have made numerous statistical comparisons which increases the chance of a type 1 error. Please apply a correction factor and explain it clearly in your work to reduce this.

4. Times of observations. Please explain clearly the time and conditions in which observations were conducted. Were observers visible to the naked mole rats?

5. Implications. It looks like each naked mole rat was tested over a single week. Was this the case? If so, a single week is not really sufficient to draw meaningful conclusions about the temporal stability of animal personality. This point needs to be considered clearly in your work, highlighting how future researchers could develop studies in this area. 

At current however, there seem to be some large revisions required in the manuscript to ensure the work is scientifically robust. I have attached the PDF version of the manuscript with specific comments. Additionally, please consider the following points: 

Round 2

Reviewer 2 Report

Dear Authors,

Thank you for submitting the revised version of this paper. It is clear that you have have addressed all of my concerns and the manuscript is now in a stronger position overall.